# Computational Approach for Probing Redox Potential for Iron-Sulfur Clusters in Photosystem I

**DOI:** 10.3390/biology11030362

**Published:** 2022-02-24

**Authors:** Fedaa Ali, Medhat W. Shafaa, Muhamed Amin

**Affiliations:** 1Medical Biophysics Division, Department of Physics, Faculty of Science, Helwan University, Cairo 11795, Egypt; fali5@vols.utk.edu (F.A.); shafaa@science.helwan.edu.eg (M.W.S.); 2Genome Science and Technology, The University of Tennessee, Knoxville, TN 37996, USA; 3Department of Sciences, University College Groningen, University of Groningen, Hoendiepskade 23/24, 9718 BG Groningen, The Netherlands; 4Universiteit Groningen Biomolecular Sciences and Biotechnology Institute, University of Groningen, 9718 BG Groningen, The Netherlands; 5Department of Physics, City College of New York, City University of New York, New York, NY 10031, USA

**Keywords:** photosystem I, iron–sulfur cluster, continuum electrostatics, broken symmetry DFT, electron transfer, MCCE

## Abstract

**Simple Summary:**

Many biological systems contain iron–sulfur clusters, which are typically found as components of electron transport proteins. Continuum electrostatic calculations were used to investigate the effect of protein environment on the redox properties of the three iron–sulfur clusters in the cyanobacterial photosystem I. Our results show a good correlation between the estimated and the measured reduction potential. Moreover, the results indicate that the low potential of F_X_ is shown to be due to the interactions with the surrounding residues and ligating sulfurs. Our results will help in understanding the electron transfer reaction in photosystem I.

**Abstract:**

Photosystem I is a light-driven electron transfer device. Available X-ray crystal structure from Thermosynechococcus elongatus showed that electron transfer pathways consist of two nearly symmetric branches of cofactors converging at the first iron–sulfur cluster F_X_, which is followed by two terminal iron–sulfur clusters F_A_ and F_B_. Experiments have shown that F_X_ has lower oxidation potential than F_A_ and F_B_, which facilitates the electron transfer reaction. Here, we use density functional theory and Multi-Conformer Continuum Electrostatics to explain the differences in the midpoint $E_{m}$ potentials of the F_X_, F_A_ and F_B_ clusters. Our calculations show that F_X_ has the lowest oxidation potential compared to F_A_ and F_B_ due to strong pairwise electrostatic interactions with surrounding residues. These interactions are shown to be dominated by the bridging sulfurs and cysteine ligands, which may be attributed to the shorter average bond distances between the oxidized Fe ion and ligating sulfurs for F_X_ compared to F_A_ and F_B_. Moreover, the electrostatic repulsion between the 4Fe-4S clusters and the positive potential of the backbone atoms is lowest for F_X_ compared to both F_A_ and F_B._ These results agree with the experimental measurements from the redox titrations of low-temperature EPR signals and of room temperature recombination kinetics.

## 1. Introduction

The photosynthesis process is the process that guarantees the existence of our life. In photosynthesis, solar energy is harvested by pigments associated with the photosynthetic machinery and stored as energy-rich compounds [1]. Initial energy conversion reactions take place in special protein complexes known as Type I and Type II reaction centers [2], which are classified according to the type of terminal electron acceptor used, iron–sulfur clusters (Fe-S) and mobile quinine for type I and type II, respectively [2,3,4,5,6,7]. Photosystem I (PS I) is the Type I reaction center found in the thylakoid membranes of chloroplasts and cyanobacteria [6,8]. PS I is a very interesting electron transfer machine that converts solar energy to a reducing power with a quantum yield close to 1 [9,10,11]. It, mainly, mediates the transfer of electrons from either cytochrome c6 or plastocyanin to the terminal electron acceptor at its stromal side through a series of redox reactions along electron transfer chains. The crystal structure of a trimeric cyanobacterial PS I is resolved at an atomic resolution of 2.5 Å [12], where each monomer consists of about 12 polypeptide chains (PSaA–PsaX) (Figure 1a,b). There are three highly conserved chains in PS I: PsaA, PsaB and PsaC [13]. The first two chains form the heterodimeric core, which noncovalently bonds most of the antenna pigments, redox cofactors employed in the electron transfer chains (ETCs) and the inter-polypeptide iron–sulfur cluster F_X_ [14,15]. PsaC comprises two iron–sulfur clusters F_A_ and F_B_, and it forms, with PsaE and PsaD, the stromal hump providing a docking site for soluble protein ferredoxin [16,17] (Figure 1a). Cofactors employed in the ETCs are a chlorophyll (a) dimer P700, two pair of chlorophyll a molecules A/A_0_ and two phylloquinones A_1_. These cofactors are arranged in two nearly symmetric branches A and B, from P700 at the luminal side to F_X_ at the PsaA and PsaB interface followed by the two terminal iron–sulfur clusters F_A_ and F_B_ (Figure 1c) [8,18,19].

Upon photoexcitation of a primary electron donor P700, an electron will transfer to the primary electron acceptor A/A_0_, within ~100 fs [20], followed by an electron transfer to the phylloquinone molecule within 20–50 ps [19]. Then, the electron is transferred, sequentially, to the three iron–sulfur clusters, F_X_, F_A_ and F_B_, within ~1.2 μs [19]. It was shown that the reduced F_B_ will directly reduce a soluble protein ferredoxin (Fd), which in turn will reduce the NADP+ to NADPH in the ferredoxin–NADP+ reductase complex (FNR) [3,4,5,6,7,21,22,23]. Knowing the redox potentials of these cofactors is crucial for understanding the primary photosynthetic processes. However, the complexity of the PS I protein complex and the electrostatic nature of interactions between charged groups and among redox centers make it difficult to assign the measured signals to a specific redox-active center. Thus, computational methods could be a complementary technique for the characterization of redox reactions. The three iron–sulfur clusters in PS I are 4Fe-4S clusters; 4Fe-4S is a distorted cube of four iron atoms linked by four bridging sulfur atoms and ligated by four cysteine ligands [8]. The PsaC polypeptide chain provides the cysteine ligands to both clusters F_A_ and F_B_: C53, C50, C20 and C47 for F_A_ and C10, C57, C13 and C16 for F_B_. The F_X_ cluster is ligated by four cysteines: two from the PsaA chain (C578 and C587) and two from the PsaB chain (C565 and C574). They are mainly distinguished by their low-temperature EPR spectrum [24,25]. In PS I, F_X_, F_A_ and F_B_ are known as low-potential [4Fe-4S] clusters that employ the 2^+^/1^+^ redox couple [26,27,28]. In its oxidized state, a low-potential [4Fe-4S] cluster has two ferric and two ferrous Fe atoms and possesses a total spin S = 0. In its reduced state, it has one ferric and three ferrous Fe atoms with total spin S = 1/2. This is due to the paramagnetic pairing between an equal-valence pair Fe^+2^−Fe^+2^ and a mixed-valence pair Fe^+2.5^−Fe^+2.5^ [8]. In PS I, the redox potentials of 4Fe-4S clusters vary in a wide range from −730 to −440 mV [19]. Low-temperature electron paramagnetic resonance (EPR) spectroscopy studies showed that the midpoint potentials are −705 ± 15, −530 and −580 mV for F_X_, F_A_ and F_B_ clusters, respectively [8,19]. However, other studies suggested that the midpoint potentials of these clusters would be positively shifted at room temperature [29,30,31,32]. Redox potentials of iron–sulfur clusters in PS I were calculated previously by Torres et al. [33]. They reported the $E_{m}$ values for F_X_, F_A_ and F_B_ to be −980, −510, and −710 mV, respectively, where the $E_{m}^{sol}$ was obtained by correcting the ionization potential calculated by gas-phase DFT with the solvation effects and referencing the calculated potential to the standard hydrogen electrode (∆SHE=−4.5 eV). Torres et al. employed a model with three dielectric regions, the continuum solvent ($\varepsilon_{wat}=80$), the protein ($\varepsilon_{prot}=4$) and ε=1 for the redox site to reflect the little screening effect of protein due to hydrogen bonding in the vicinity of the clusters. In their paper, Ptushenko et al. [34] argued the implausibility of the proposed three dielectric regions model by Torres due to the overestimation of the amide field in the vicinity of clusters, which lead to a negatively deviated midpoint potential from experimental values by 275 to 330 mV for F_X_ and 130 to 245 mV for F_B_. In the work of Ptushenko and coworkers [34], they calculated midpoint potentials for all redox cofactors in PS I, including the three [4Fe-4S] clusters. Their reported values are −654, −481 and −585 mV for F_X_, F_A_ and F_B,_ respectively. In their calculations, they employed the semicontinuum electrostatic model, where two dielectric constants for proteins were used, the optical dielectric constant ($\varepsilon_{o}=2.5$) for pre-existing permanent charges and a static dielectric constant $(\varepsilon_{s}=4)\;$ for charges formed due to the formation of ions in protein upon ionization reaction. In their calculations, Torres and Ptushenko included all protein subunits and other prosthetic groups in the PS I complex.

Here, we report the calculated relative midpoint potential of [4Fe-4S] clusters F_X_, F_A_ and F_B_, using Multi-Conformer Continuum Electrostatics (MCCE) [35,36,37]. Although this model is based on classical electrostatics and does not consider the quantum effects such as spin–spin and spin–orbit interactions, it successfully produced the experimental pK_a_ and *E_m_* values for oxo-manganese complexes and Mn and Fe superoxide dismutase [38,39]. Moreover, we provide an insight into the conformational changes and the interactions that induce the differences in the redox potential of the three [4Fe-4S] clusters from the classical electrostatics perspective and their implication on the electron transfer reaction.

## 2. Materials and Methods

### 2.1. Structural Model

PS I crystal structure of Thermosynechococcus elongatus (PDB code: 1JB0 [12]), at resolution 2.5 Å, was used as the basis for all calculations. All crystallographic water molecules and lipids were removed prior to our calculations. There are 49 cofactors and 35 amino acids with missing atoms. Moreover, atomic coordinates of ~91 residues were not reported during experiments. Prior to MC calculations, PDBfixer (Python-based software) [40] was used to add missed inter-polypeptide residues only and to add all other missed atoms in residues and cofactors.

The fixed structure was solvated by water molecules (more than 200,000) to fill a cubic box (~19 × 19 × 19 Å). Sodium counterions were added by replacing water molecules to provide a neutral simulation box. The resulting complex comprised approximately 700,000 atoms. Atomic interactions for the standard amino acids and cofactors (except 4Fe-4S clusters) in PS I were based on the developed AMBER molecular potentials for PS II [41]. The parameterization of the iron–sulfur clusters was based on the work of Carvalho et al. [42]. Energy minimization calculation with the steepest descent method followed by conjugate gradient energy minimization was conducted using the Gromacs program. The minimized structure was further equilibrated for 100 ps in the NVT ensemble followed by another NPT equilibration for 100 ps. Then, a short MD simulation (10 ns) was performed, where trajectories were collected every 10 ps. The final coordinates retrieved from MD simulation were used for further calculations. On the other hand, the geometry for [4Fe-4S] clusters F_X_, F_A_ and F_B_, surrounded by ~10 Å nearby residues, as shown in Figure 2, were extracted from the crystal structure and were optimized at the DFT/B3LYP level of theory, with LANL2DZ basis sets [43] for Fe metal centers and 6–31G* basis set for other atoms, using the Gaussian09 package [44]. The [4Fe-4S] core was set to the reduced state with total spin S = ½ using the broken symmetry wavefunction [45]. Each DFT optimized structure was docked into the complete PS I structure retrieved from MD simulations such that the coordinates of side chains that are not included in the DFT clusters and all backbone atoms remain the same as in MD-obtained structures. This final complex was used as an initial state for MC simulations (structures are available in Appendix A). 

### 2.2. Multi-Conformer Continuum Electrostatics (MCCE) Calculations

MCCE generates different conformers for all amino acid residues and cofactors. These conformers undergo a preselection process, which discards conformers that experience vdW clashes [36]. All crystallographic water molecules and solvated ions are stripped off and replaced with a continuum dielectric medium. The electrostatic potential of the protein is calculated by solving Poisson–Boltzmann equation [46] using DelPhi [47]. In this calculation, the surrounding solvent (water) was assigned a dielectric constant of ε=80, and ε=4 was assigned for protein [48]. The partial charges and radii used for amino acids in MCCE calculations are taken from the PARSE charges [49]. The probe radius for placing water is 1.4 Å and 0.15 M salt concentration is used. For 4Fe-4S clusters, each Fe ion, bridging sulfurs S and each ligand as separate fragments with an integer charge interact with each other through electrostatic and Lennard–Jones potentials [39]. 

The Fe atoms have formal charges of +2 or +3, while each bridging sulfur atom has a charge of −2. 

For each conformer i, DelPhi calculates different energy terms, the polar interaction energy ($∆G_{pol,i}$), desolvation energy $∆∆G_{rxn,i}$ and pairwise electrostatic and Lennard–Jones interactions with other conformers *j* ($∆G_{ij}$). For M conformers, the $∆∆G_{rxn,i}$ and $∆G_{pol,i}$ energy terms will be collected into two matrices with M rows while the $∆G_{ij}$ energy term will be collected into the M × M matrix [37]. A single protein microstate x is defined by choosing one conformer for each cofactor and residue. Therefore, the number of possible microstates of the system is very high. As a final step, MCCE uses Monte Carlo sampling to compute the probability of occurrence for each conformer in the Boltzmann distribution for a given pH and electron concentration ($E_{h}$) [37,50].

The total energy of each microstate G_x_ with M conformers is the sum of electrostatic and non-electrostatic energies, and it is computed according to the following equation [51,52]:(1)$$∆G_{x}=\overset{M}{\underset{i=1}{\sum }}\delta_{x,i}\left[\left(2.3m_{i}k_{b}T\left(pH-pK_{sol,i}\right)+n_{i}F\left(E_{h}-E_{m,sol,i}\right)\right)+\left(∆∆G_{rxn,i}+∆G_{pol,i}\right)+\overset{M}{\underset{j\neq i}{\sum }}\delta_{x,j}∆G_{ij}\right]\;$$
where $\delta_{x,i}$ is equal to 0 if microstate x lacks conformer i and 1 otherwise; $m_{i}\;$ takes the values 0, 1 and −1 for neutral, base and acid conformers, respectively; $n_{i}$ is the number of electrons transferred during redox reactions; $pK_{a,sol,i}$ and $E_{m,sol,i}$ are the reference $pK_{a}$ and $E_{m}$ for *i*-th group in the reference dielectric medium (e.g., water); F is the Faraday constant; K_b_ is the Boltzmann constant; T is temperature (298 K in our calculations); $∆∆G_{rxn,i}$ is the desolvation energy of moving conformer *i* from solution to its position in the protein; $∆G_{ij}$ is the pairwise interaction between different conformers *i* and *j*; and $∆G_{pol,i}$ is the pairwise interaction of conformer *i* with other groups with zero conformational degrees of freedom (e.g., backbone atoms). 

The reference solution $E_{m,sol}$ for Fe ions is obtained according to the thermodynamic cycles shown in Figure 3. The solution redox potential, *E_m,sol_* is fixed at −170 mV, a value that reproduced the redox chemistry of F_X_ (−715 mV vs. SHE) [39].

### 2.3. Mean-Field Energy (MFE) Analysis

MCCE determines the in situ midpoint potential $E_{m}$ of the redox centers as shifted by the protein environment. This shift is due to the loss in the reaction field energy $∆∆G_{rxn}$ and other electrostatic interactions. Mean-field energy analysis (MFE) allows decomposition of these energetic terms to determine what factors yield the reported midpoint potentials in protein, Equation (2) [53],
(2)$$FE_{m,MFE}=nFE_{m,sol}+∆G_{bkbn}+∆∆G_{rxn}+∆G_{res}^{MFE}\;\;\;\;\;\;\;$$
where $∆G_{bkbn}$ is the electrostatic and non-electrostatic interactions of the redox cofactor with the backbone atoms of protein and $∆G_{res}^{MFE}$ is the mean-field electrostatic interaction between the redox cofactor and the average occupancy of conformers of all other residues in the protein in the Boltzmann distribution at each $E_{h}$ [53]. Other terms are the same as shown in Equation (1).

## 3. Results and Discussion

Molecular structures for [4Fe-4S] clusters in PS I were investigated by extended X-ray absorption fine structure (EXAFS), which revealed two peaks at ~2.27 and ~2.7 Å, which are attributed to the backscattering from sulfur and iron atoms, respectively [54,55,56,57]. The results of geometry optimization of three extracted structures with total spin S = ½ and with [4Fe-4S] in their reduced state are reported in Table 1a. Our calculated Fe-S (bridging sulfur atoms), Fe-SG (organic sulfur atoms) and Fe-Fe bond distances are shown, generally, to be longer than the XRD [12] and EXAFS reported distances (Table 1).

**Table 1 biology-11-00362-t001:** Fe-S and Fe-Fe bond distances in 4Fe-4S Clusters from XRD, EXAFS experiments and DFT geometry optimization.

	DFT			XRD		
	F_X_	F_A_	F_B_	F_X_	F_A_	F_B_
Fe-S (Å)	2.32	2.32	2.34	2.3 (×1)	2.3 (×7)	2.3 (×12)
	2.35	2.37	2.37	2.2 (×1)	2.2 (×4)	
	2.39	2.38	2.38		2.4 (×1)	
	2.45	2.39	2.4			
	**2.46**	2.4	2.4			
	2.47	2.44	2.4			
	2.49	2.46	2.44			
	2.52	**2.46**	**2.44**			
	2.52	2.47	**2.49**			
	**2.44**	**2.48**	2.49			
	**2.44**	**2.52**	**2.5**			
	2.44	2.57	2.53			
Fe-SG (Å)	2.36	2.49	2.39	2.4 (×2)	2.4 (×1)	2.4 (×2)
	2.37	2.35	2.4	2.2 (×1)	2.3 (×1)	2.3 (×2)
	**2.35**	**2.34**	**2.35**	2.3 (×1)		
	2.34	2.34	2.36			
**Avg.**	**2.42**	**2.45**	**2.45**			
Fe-Fe (Å)	2.96	3.16	3.05	2.7 (×6)	2.7 (×4)	2.7 (×6)
	2.97	3.02	3.14			
	3.04	3.19	2.83			
	3.18	2.95	3.15			
	3.29	3.2	3.1			
	3.15	2.97	3.02			
**EXAFS (Å)**						
**Fe-S**			Fe-Fe			
2.27			2.7			

Bold values are the distances between the second oxidized Fe ion and the four sulfur ligands (three bridging sulfurs and one sulfur from cysteine), while Avg. is the average over these distances for each 4Fe-4S cluster (see Figure 4).

### The Midpoint Potentials $\left(E_{m}\right)$ of F_X_, F_B_ and F_A_

In our calculations, we considered the oxidation potential of the second oxidized Fe ion as the oxidation potential of the cluster from [4Fe-4S]^+1^ to [4Fe-4S]^+2^. The measured $E_{m}$ values of F_X_, F_A_ and F_B_ are reported in Table 2. For F_X_, $E_{m}$ is −715 mV, which is ~0–65 mV more negative than experimental values [8,58,59]. The $E_{m}$ values calculated for F_A_ and F_B_ are −355 and −346 mV, respectively, and are positively shifted by more than 80 mV compared to experimentally determined values [19,60,61]. Our results are shown to agree with the experimental values within the error range of the method [37,62,63]. To better understand the effect of ligands and other residues in the model structures on the calculated $E_{m}$, mean-field energy (MFE) analysis was performed for each 4Fe-4S cluster at its calculated $E_{m}$ to determine the different factors contributing to the stabilization of the ionization state of clusters in the protein (Equation (2)). Results from the MFE analysis are reported in Table 3. The calculated $E_{m}$ values are shown to deviate from the reference value by about 13 kcal/mol for F_X_, while for both F_A_ and F_B_ it is shown that the values are only shifted by about 4 kcal/mol, which may be attributed to the degree of burial of each 4Fe-4S cluster. Our calculations demonstrated that the desolvation energy $∆∆G_{rxn}$ was always positive and unfavorable. It destabilizes the ionization state by ~80 kcal/mol for F_X_ and by ~76 kcal/mol for F_A_ and F_B_. This unfavorable interaction is, nearly, compensated by the electrostatic interactions with the surrounding residues $∆G_{\mathrm{resd}}$ for F_X_, F_A_ and F_B_ to be −104, −97 and −92 kcal/mol, respectively.

Moreover, interactions with the backbone $∆G_{\mathrm{bkbn}}$ disfavor the oxidized form of the Fe. For F_A_ and F_B,_ this effect is shown to be ~1.13-fold more than that in F_X_. By further breaking down the contribution from different residues and ligands (see Table 4), it is shown that stabilization of ionization state of 4Fe-4S clusters is mainly controlled by the classical electrostatic interactions between Fe ions and both bridging sulfurs and cysteine ligands. Oppositely, the electrostatic interaction with positively charged residues and other Fe ions is shown to destabilize the oxidized form of Fe ion. 

Furthermore, to investigate the role of different interactions in altering the redox potential of 4Fe-4S clusters, interaction energies between bridging sulfurs and the average occupancy of conformers of all other residues in the protein complex were calculated at the $E_{m}$ of each cluster. The total interaction energy of S3 in the F_B_ cluster with other residues and cofactors in PS I protein is shown to be the strongest interaction compared to both S3 in F_X_ and F_A_ and compared to other sulfur atoms in the same cluster (Figure 5b). By further breaking down the contribution from different residues, it is shown that interaction energies are favorable between inorganic sulfurs and the charged and polar amino acids in the vicinity of the clusters. The electrostatic interaction between Lys-C51 and S2 of F_A_ clusters is shown to be the strongest interaction compared to other bridging sulfurs in the same cluster or in the other two clusters with interaction energy ~17 kcal/mol (see Table 5 and Figure 5a). The bridging sulfurs in the inter-polypeptide cluster F_X_ are hydrogen-bonded to the charged residues, mainly Lys and Arg residues, in the vicinity of the cluster with a total energy of ~−152.5 kcal/mol. However, the stromal clusters exhibited less interaction between bridging sulfurs and polar and charged residues, where the total interaction energies with inorganic sulfurs in F_A_ and F_B_ are −126 and −30 kcal/mol, respectively.

## 4. Conclusions

Based on our calculations, the low potential of F_X_ is shown to be due to the backbone and residue sidechain contributions. Moreover, the distances between ligating sulfurs and the second oxidized Fe ion were found to be, on average, ~2.3 Å for F_X_ and ~2.5 Å for F_A_ and F_B_. This could explain the stronger effect of sulfurs in F_X_ for shifting the redox potential. Moreover, the bridging sulfurs in the inter-polypeptide cluster F_X_ interact with the charged residues in the cluster’s proximity, namely Lys and Arg residues, with a total interaction energy of −152.5 kcal/mol. The stromal clusters, on the other hand, showed reduced contact between bridging sulfurs and polar and charged residues, with total interaction energies with inorganic sulfurs of −126 and −30 kcal/mol in F_A_ and F_B_, respectively. When compared to S3 in F_X_ and F_A_, as well as other sulfur atoms in the same cluster, the overall interaction energy of S3 in the F_B_ cluster with other residues and cofactors in PS I protein is demonstrated to be the greatest. Finally, when compared to other bridging sulfurs in the same cluster or in the other two clusters, the electrostatic interaction between Lys-C51 and S2 of F_A_ clusters is revealed to be the strongest interaction, with an interaction energy of 17 kcal/mol. In this study, our calculations demonstrated how each 4Fe-4S cluster is exposed to a different protein electrostatic environment, which tunes the redox properties of each 4Fe-4S cluster.

## Figures and Tables

**Figure 1 biology-11-00362-f001:**
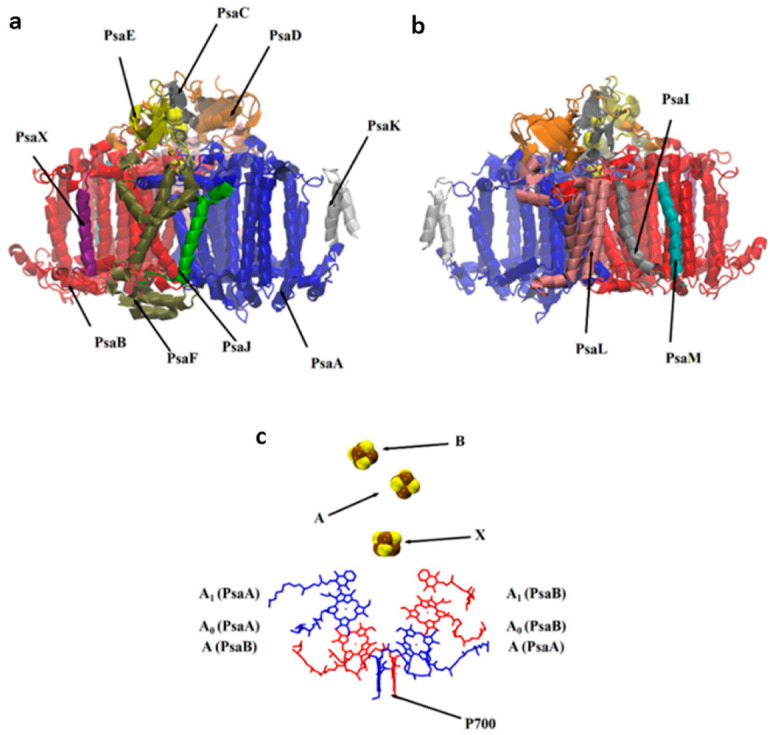
PDB code 1JB0 [12]: Twelve protein subunits in the polypeptide structure of cyanobacterial PS I monomer viewed perpendicular to the plane of the thylakoid membranes. (**a**) Front view; (**b**) back view; (**c**) electron transfer chains (ETCs) in PS I, including primary electron donor P700 (Chl a dimer), primary electron acceptors A/A_0_ (Chl a molecules), secondary electron acceptor A_1_ (phylloquinone molecule PQN), tertiary electron acceptor X (F_X_) and terminal electron acceptor A (F_A_) and B (F_B_) [8].

**Figure 2 biology-11-00362-f002:**
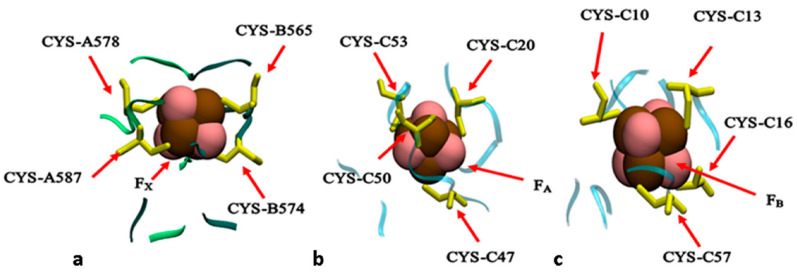
Structural models used in this study. (**a**–**c**) The iron–sulfur clusters in PS I surrounded by nearby amino acids (~10 Å) from PsaA/PsaB and PsaC subunits. The letters A, B and C refer to the subunits PsaA, PsaB and PsaC, respectively. (**a**) The inter-polypeptide 4Fe-4S cluster F_X_ and the surrounding amino acids from both protein domains PsaA and PsaB. (**b**,**c**) The stromal iron–sulfur clusters F_A_ and F_B,_ respectively, surrounded by near residues from PsaC subunit.

**Figure 3 biology-11-00362-f003:**
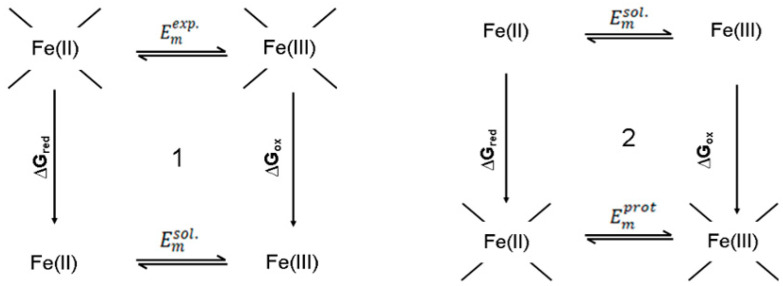
Thermodynamic cycle for the redox reaction ${\mathrm{Fe}}^{+2}⇌{\;\mathrm{Fe}}^{+3}+e^{-1}$, where $E_{m}^{exp}$ is the midpoint potential determined in experiment, $E_{m}^{sol}$ is the midpoint potential in reference medium and $E_{m}^{prot}$ is the midpoint potential in situ calculated by MCCE.

**Figure 4 biology-11-00362-f004:**
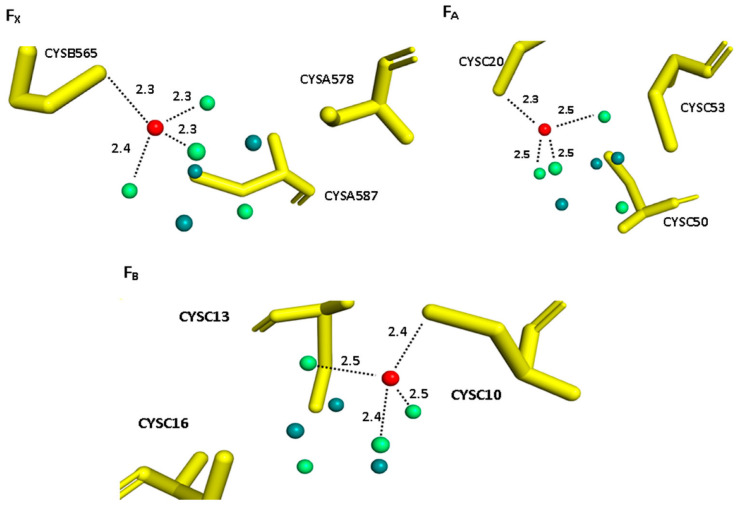
The structure of optimized F_X_, F_A_ and F_B_ redox centers, showing distances between the second oxidized Fe ion and the four ligating sulfurs (three bridging sulfurs and one sulfur from cysteine). Red spheres are the second oxidized Fe ion, deep teal spheres are the other Fe ions in the cluster, lime green spheres are bridging sulfurs and yellow sticks are cysteine ligands.

**Figure 5 biology-11-00362-f005:**
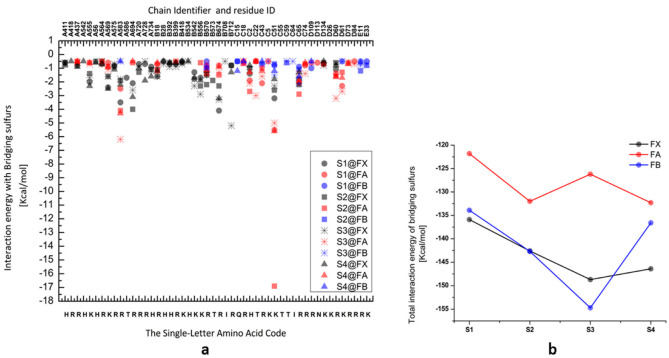
The interaction energy of bridging sulfurs with the surrounding residues. (**a**) The selected interaction energies of the bridging sulfurs with different residues. (**b**) The total interaction energy of bridging sulfurs in each 4Fe-4S cluster with the surrounding residues and other cofactors in the PS I protein complex.

**Table 2 biology-11-00362-t002:** Calculated midpoint potential for redox couples +2/+1 (in units of mV).

	Cal. *E_m_*	Exp. *E_m_*
F_A_	−355	−440 ^j^, −53 0 ^i^, −50 0^m^
F_B_	−346	−465 ^j^, −580 ^i^, −550 ^m^
**F_X_**	**−715**	**−650 ^m^, −705 ^k^, −670 ^l^**

The bold is the $E_{m}$ value used for determining the solution reference redox potential; ^i^ [24,64], ^j^ [61], ^k^ [58], ^l^ [31], ^m^ [65].

**Table 3 biology-11-00362-t003:** Energy terms that contribute to the shift of the redox potential in protein. These terms are shown to be the desolvation energy term $∆∆G_{rxn}$, backbone contribution $∆G_{\mathrm{bkbn}}$ and pairwise interaction with sidechains $∆G_{\mathrm{resd}}$ (energies are in units of kcal/mol).

∆G	$F_{X}$	$F_{A}$	$F_{B}$
$∆G_{\mathrm{bkbn}}$	15.10	17.03	16.61
$∆∆G_{rxn}$	79.99	75.70	76.27
$∆G_{\mathrm{resd}}$	−104.32	−96.66	−91.77
$∆E_{m}$	12.81	4.35	4.14

**Table 4 biology-11-00362-t004:** Selected electrostatic interaction between iron–sulfur clusters and the surrounding residues (energies are in units of kcal/mol) ^a^.

	F_X_		F_A_		F_B_	
	Residues	Energy	Residues	Energy	Residues	Energy
4Fe-4S	**S1@Fx**	**−57.9**	**S4@Fa**	**−49.38**	**S3@Fb**	**−53.65**
	**S2@Fx**	**−56.31**	**S3@Fa**	**−48.31**	**S1@Fb**	**−49.47**
	**S3@Fx**	**−55.31**	**S1@Fa**	**−47.33**	**S2@Fb**	**−47.19**
	**C/B565**	**−24.94**	**C/C20**	**−24.61**	**C/C10**	**−28.55**
	FE3@Fx	40.98	FE1@Fa	34.44	FE1@Fb	54.62
	FE2@Fx	45.68	FE3@Fa	53.66	FE2@Fb	42.15
	FE1@Fx	60.88	FE4@Fa	36.45	FE3@Fb	35.66
Surrounding residues	C/B574	−10.92	S2@Fa	−22.32	S4@Fb	−22.19
	C/A587	−10.04	C/C53	−13.45	C/C16	−7.92
	C/A578	−9.33	C/C47	−7.89	C/C57	−7.67
	S2@Fa	−3.01	C/C50	−7.7	C/C13	−5.56
	D/A593	−2.24	C/C16	−3.63	S1@Fa	−2.78
	S4@Fa	−1.97	S3@Fb	−3.36	S4@Fa	−2.1
	S3@Fa	−1.86	S4@Fb	−3.12	C/C53	−1.88
	E/C54	−1.83	E/D62	−2.73	S2@Fa	−1.74
	C/C50	−1.73	S1@Fb	−2.45	S3@Fa	−1.37
	E/A699	−1.46	E/C54	−2.06	E/C54	−1.08
	S1@Fa	−1.45	D/C23	−1.89	C/C20	−0.91
	E/C71	−1.29	S1@Fx	−1.74	S/C63	−0.71
	E/B679	−1.19	S2@Fb	−1.53	C/C47	−0.68
	C/C53	−0.94	S2@Fx	−1.35		
	C/C47	−0.91	C/C57	−1.21		
	E/B682	−0.79	S4@Fx	−1.08		
	D/B566	−0.77	S3@Fx	−1.07		
	E/D62	−0.73	D/C46	−1.05		
	D/B558	−0.7	C/B565	−0.81		
	D/B546	−0.68	C/A578	−0.79		
	D/B14	−0.65	E/C71	−0.77		
	S4@Fb	−0.64	C/C10	−0.75		
	S3@Fb	−0.62	E/D103	−0.68		
	D/B29	−0.5	D/B566	−0.59		
			C/A587	−0.56		
			E/B679	−0.53		
			E/B682	−0.52		
			C/C13	−0.51		

^a^ bold values are for the sulfurs bound to the 2nd oxidized Fe ion. @ Indicates to which cluster the atom belongs.

**Table 5 biology-11-00362-t005:** Selected electrostatic interactions between bridging sulfur atoms and the surrounding residues (kcal/mol).

		F_X_					F_A_					F_B_				
		S1	S2	S3	S4	Tot.*	S1	S2	S3	S4	Tot.*	S1	S2	S3	S4	Tot.*
K	C51	−3.2	−2.6	−2.3	−1.8	−9.9	−5.5	−16.9	−5	−5.6	−33	−0.8	−0.7	−1.5	−1.2	−4.2
R	A583	−3.5	−1.9	−1.8	−2.2	−9.4	−2.5	−4.1	−6.2	−4.3	−17			−0.5	−0.5	−1
R	C65	−1.8	−2.2	−1.7	−1.3	−7	−2.1	−2.9	−1.4	−1.9	−8.3	−0.9	−1.1	−2.1	−1.6	−5.7
H	C2	−1.4	−1	−1.1	−0.8	−4.3	−1.9	−2.7	−2.1	−1.3	−8					0
R	D60	−1.1	−0.6	−0.7	−0.8	−3.2	−1.4	−1.6	−3.2	−1.6	−7.8					0
R	B674	−4.1	−2.3	−3.3	−3.2	−13	−0.8	−1.5	−1.3	−0.9	−4.5					0
R	B570	−1.5	−2.2	−1.5	−1.2	−6.4	−0.9	−1.4	−0.7	−1.1	−4.1	−0.5	−0.8	−1	−1	−3.3
K	D61					0	−2.3	−1.3	−2.7	−1.7	−8	−0.5		−0.5		−1
R	B18	−1.6	−1.1	−1.6	−1.2	−5.5	−0.7	−1.2	−0.8	−0.6	−3.3					0
R	C43					0	−2.1	−1	−1.6	−1.2	−5.9	−0.7		−0.7	−0.5	−1.9
K	A569	−2.4	−1.6	−1.6	−2.5	−8.1	−0.6	−0.9	−1.1	−0.9	−3.5					0
K	D134	−0.7	−0.6	−0.8	−0.5	−2.6		−0.7			−0.7					0
R	C74					0	−0.8	−0.7	−1.4	−0.6	−3.5					0
R	A694	−2.1	−4	−2.6	−3.1	−12		−0.6		−0.5	−1.1					0
K	A555	−2	−1.4	−1.7	−2.3	−7.4		−0.6	−0.7	−0.6	−1.9					0
K	B556	−1.7	−2.3	−2.9	−1.9	−8.8		−0.6			−0.6					0
R	A437	−0.8	−0.6	−0.6	−0.9	−2.9		−0.5	−0.6	−0.5	−1.6					0
R	A564	−0.7	−0.5	−0.5	−0.7	−2.4		−0.5	−0.7		−1.2					0
R	B399	−0.7	−0.6	−0.9	−0.6	−2.8		−0.5			−0.5					0
T	C22					0	−0.5	−0.5	−3	−0.5	−4.5					0
T	A586	−1.7				−1.7					0					0
K	B542	−1.3	−1.8	−2.3	−1.7	−7.1					0					0
H	A734	−0.9	−1.1	−1.2	−1.6	−4.8					0					0
R	A575	−0.8	−0.9	−0.7	−1.1	−3.5					0					0
R	B712	−0.8	−0.8	−5.2	−1.3	−8.1					0					0
R	A720	−0.7	−1.3	−0.9	−1	−3.9					0					0
R	A728	−0.7		−0.5	−1.9	−3.1					0					0
H	B392	−0.7	−0.6	−0.9	−0.6	−2.8					0					0
H	A411	−0.6	−0.6	−0.6	−0.8	−2.6					0					0
H	B28	−0.5		−0.7	−0.5	−1.7					0					0
K	B418	−0.5	−0.6	−0.7	−0.5	−2.3					0					0
R	A418				−0.5	−0.5					0					0
H	A542				−0.5	−0.5					0					0
H	A56		−0.5		−0.6	−1.1					0					0
H	B534			−0.5	−0.5	−1					0					0
T	B573		−1.9			−1.9					0					0
I	B708			−0.5		−0.5					0					0
Q	C15					0					0	−0.5	−0.5		−1.2	−2.2
R	C18					0	−0.5		−0.5	−0.7	−1.7	−0.6			−0.5	−1.1
K	C5					0	−0.5				−0.5			−0.5		−0.5
T	C55					0					0					0
T	C59					0					0		−0.6	−0.5		−1.1
I	C64					0					0			−0.5		−0.5
R	D109					0	−0.6				−0.6	−1		−0.6	−0.5	−2.1
N	D113					0	−0.6				−0.6					0
K	D26					0					0					0
R	D73					0	−0.6		−0.6	−0.7	−1.9					0
R	D84					0	−0.5		−0.7		−1.2					0
R	E11					0					0	−0.5	−1.2	−0.8	−0.6	−3.1
K	E33					0					0	−0.5	−0.8	−0.7		−2

* total interaction energy of bridging sulfurs with each residue in the vicinity of the cluster.

## Data Availability

The structural models that were used are available in the SI online.

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
