# Peer review of "Computational Approach for Probing Redox Potential for Iron-Sulfur Clusters in Photosystem I"

_biology, 2022, doi:10.3390/biology11030362_

Round 1

Reviewer 1 Report

This paper elucidates the differences in the redox potential of iron-sulfur cluster by using density functional theory and multi-continuum electrostatics. This is a potential interesting and important study of Fe-S clusters. The data presented here provide new insight into the function of Fe-S clusters. There are, however, several concerns that needed to be addressed before publication can be considered. The authors should discuss the following points.

  1. Given the success of simple analysis, the methodology will be extended the study of other Fe-S clusters.
  2. Hydrogen bonding interactions and amide dipole with Fe-S clusters may play a key role in modulating the redox potential. 

Author Response

  1. Given the success of simple analysis, the methodology will be extended the study of other Fe-S clusters.

This method has been used previously with the Mn cluster of the oxygen evolving complex of PSII and superoxide dismutase (please see reference 38 and 39). Using this simple analysis to study other Fe-S cluster would be useful to investigate how protein environment tunes the redox potential of different iron-sulfur clusters (like Complex I). This also could further support the usefulness of the method.

  1. Hydrogen bonding interactions and amide dipole with Fe-S clusters may play a key role in modulating the redox potential. 

We calculated the interaction energies between bridging sulfurs and residues in the vicinity of the clusters, we added a table (table 5) and two figures (figure 5a, 5b), which are  showing these interactions.

Reviewer 2 Report

The paper by Amin and coauthors describes quantum chemical calculations of redox potentiasl for iron-sulfur clusters taking part in photosynthesis process. This goal is quite important for understanding a complex mechanism of photosynthesis. The authors did a reasonable simplification of the structure of the studied complex iron-sulfur clusters in protein subunits. As results, energies of different conformres, geometry of Fe-S clusters,and their midpoint redo[ potentials were calculated.  The authors are trying to explain the difference between experimental and calculated values of potentials in terms of geometry of Fe-S clusters.

The main comments on the paper is related to the absence of a conclusion paper. Please, summarize the data obtained in a special conclusion section!

Other remarks as follows.

It seems there is a gap in a numbering of references. In page 3, ref. no. 62 goes after ref. no. 32!

In page 3, line 9, on the right, it should be: "...to -440 mV...", "mV" was missing.

Author Response

  1. The main comments on the paper is related to the absence of a conclusion paper. Please, summarize the data obtained in a special conclusion section!

We added a conclusion section after the last table.

  1. It seems there is a gap in a numbering of references. In page 3, ref. no. 62 goes after ref. no. 32!

We revised and corrected the references in the manuscript

  1. In page 3, line 9, on the right, it should be: "...to -440 mV...", "mV" was missing.

Checked and corrected